# Craniomaxillofacial Fibrous Dysplasia Improved Cosmetic and Occlusal Problem by Comprehensive Treatment: A Case Report and Review of Current Treatments

**DOI:** 10.3390/diagnostics12092146

**Published:** 2022-09-03

**Authors:** Kisho Ono, Norie Yoshioka, Yuki Kunisada, Tomoya Nakamura, Yuko Nakamura, Kyoichi Obata, Soichiro Ibaragi, Shogo Minagi, Akira Sasaki

**Affiliations:** 1Department of Oral and Maxillofacial Surgery, Graduate School of Medicine, Dentistry and Pharmaceutical Sciences, Okayama University, Okayama 700-8525, Japan; 2Department of Occlusal and Oral Functional Rehabilitation, Graduate School of Medicine, Dentistry and Pharmaceutical Sciences, Okayama University, Okayama 700-8525, Japan

**Keywords:** fibrous dysplasia, polyostotic, craniomaxillofacial, surgical, prosthetic, comprehensive treatment

## Abstract

Fibrous dysplasia (FD) is a fibrous lesion of immature bone, with an incidence of 10–20% in the head and neck region. Most cases are monostotic, but when a lesion occurs on the maxillofacial region and spreads to the surrounding bone, it is classified as polyostotic, despite its localized occurrence. In some cases, surgical intervention is required to improve the cosmetic or functional disturbance of a FD in the maxillofacial region, but it is necessary to confirm symmetry of the maxillofacial region in real time, and a surgical support system is required to compensate. Furthermore, prosthetic intervention is considered when postoperative acquired defects occur or further cosmetic or occlusal function improvement is needed. A comprehensive approach by an oral surgeon and a maxillofacial prosthodontist is necessary for the successful treatment and rehabilitation of such patients. In this article, we describe the case of a craniomaxillofacial FD patient with facial asymmetry and denture incompatibility with improved quality of life measures by integrating surgical treatment using a navigation system and postoperative prosthetic rehabilitation. We also discuss recent diagnostic methods and treatment strategies for craniomaxillofacial FD in the literature.

## 1. Introduction

Fibrous dysplasia (FD) is a non-neoplastic bone lesion in which normal bone tissue is replaced by immature bone tissue and fibrous connective tissue. FD is more common in children and young adults and is more common in women than men [1,2]. FD is generally classified into two categories: monostotic type or polyostotic type [3,4,5]. The most common sites of FD are the long canal bone, craniomaxillofacial bone, and ribs. When FD occurs in the maxillofacial bones, osteotomy or debridement is performed for facial asymmetry. However, there is still no consensus on the best way to solve the problem. In addition to surgical approaches, postoperative quality of life (QOL) improvement through occlusal reconstruction is another issue that needs to be addressed.

The present case report aimed to describe the successful outcome of a craniomaxillofacial FD treated with comprehensive treatment including surgery and prosthetic treatment for facial deformity and denture incompatibility. This study was reported in line with the CARE Guidelines [6].

## 2. Case Presentation

A 63-year-old woman was referred by her dentist to our department in October 2018 with masticatory disorders due to facial asymmetry and denture incompatibility. She had a history of left-sided frontal and parietal meningoceles, left-sided hearing loss, and cervical spondylosis. Physical examination showed horizontal tilting of the maxillary occlusal plane up to the right and facial asymmetry with right-sided deviation of the mandibular midline at the first visit (Figure 1A). The denture used was a crossbite arrangement aligned on the right side (Figure 1B), with no associated opening disorder. Computed tomography (CT) imaging showed slit-glass-like bone thickening from the left temporal area to the base of the skull and zygomatic bone (Figure 2A,B). In addition, there was bone proliferation noted in the left alveolar region on the maxilla and enlargement of the left articular head on the mandible (Figure 2C,D). Magnetic resonance (MR) imaging revealed irreducible anterior disc displacement of bilateral articular disks and joint effusion in the joint capsule on the left side (Figure 3). Based on these findings, a diagnosis of FD was assigned.

We planned to remove the hyperplastic bone of the maxillary alveolar, zygomatic, and articular head to improve the facial asymmetry and the occlusal plane and to fabricate a denture with the newly obtained occlusal plane to restore the masticatory function. In surgery, the extent of bone removal was estimated intraoperatively using a navigation system (Brainlab AG, Munich, Germany) in order to remove the bulging bone so that facial symmetry could be obtained (Figure 4A,B). In the resection of the articular head, the joint capsule was separated; the upper joint cavity was opened; and the excess articular head was resected (Figure 4C,D). After confirming that there was sufficient mobility of the mandible, the excess articular tuberosity was removed, and subcutaneous fat was harvested from the abdomen and grafted between the resected articular head and the articular disc (Figure 4E). The removed hard tissue was submitted for histopathology (Figure 5).

Postoperatively, the patient began to practice opening and closing the mouth on the sixth day, and the original denture was repaired and fitted on the 14th day. Subsequently, the maxillofacial prosthodontist adjusted the denture accordingly, and the jaw position was stabilized one year after surgery (Figure 6A,B). Although some facial asymmetry remains, an overlay of preoperative and postoperative three-dimensional (3D) craniomaxillofacial bone models demonstrates a favorable improvement in jaw deviation (Figure 7). The masticatory disorder has also significantly improved, and the patient is satisfied with the results. At present, there is no evidence of bone regrowth, and the patient is progressing well.

## 3. Discussion

FD is classified into monostotic fibrous dysplasia (MFD) and polyostotic fibrous dysplasia (PFD), with some PFDs including the following: Albright’s syndrome, with skin pigmentation and endocrine disturbances resulting in premature sexual maturation in females, and Jaffe’s type, with mild polyostotic fibrous dysplasia and melanin pigmentation in the skin around the bone lesions [7,8]. In the literature, it is reported that PFD accounts for approximately 15–20% of all FD cases, while Albright syndrome accounts for ~3% of PFD. One problem with the classification of MFD and PFD alone is that the craniomaxillofacial bones where FD predominantly occurs, when it occurs at the sutures of the facial skeleton, are considered to be two regions, resulting in a diagnosis of PFD. Thoma proposed that the facial bones be considered as a single region with the condition similar to MFD, and the diagnosis of craniomaxillofacial FD should be unified [9]. Our case also shows thickening of the maxilla, mandible, zygomatic bone, and temporal bone on imaging examination; it is classified as PFD according to the existing classification. However, if the classification proposed by Thoma is used, this case can also be considered a case of craniomaxillofacial FD [9,10]. The most common age of onset is in the 20s, but, due to the fact that the disease progresses with bone growth, it is estimated that about 80% of cases occur before the age of 20. In terms of staging and treatment, Barrionuevo et al. classified the disease as stage I (latent stage), stage II (symptomatic stage), and stage III (complicated stage) and stated that stage I should be followed up, while stage II should be treated with surgery [11]. From another perspective, radiotherapy is contraindicated because it may cause malignant transformation of FD to sarcoma [12]. It has been reported that malignant transformation occurs in 0.4% of patients undergoing follow-up without radiotherapy, and so malignant transformation should be considered if there is pain in the swollen area, rapid growth of the neoplastic lesions, or an increase in alkaline phosphatase (ALP) level [12,13]. Although our patient did not have an elevated ALP level, we will continue to follow up the patient carefully and adequately due to the long course of her original disease.

Imaging studies are useful in the diagnosis of FD. Leeds et al. classified imaging diagnosis into three categories from the simple X-ray images of the craniomaxillofacial FD: (i) pagetoid form, which is the most common type, showing prominent thickening of the crown of the skull and a mixture of translucent and sclerotic images inside, similar to Paget’s disease; (ii) sclerotic form, which shows a hyperostotic image described as a frosted glass image; and (iii) cystic form, which is often found in the cranial corona and shows unilocular or multilocular translucency with thick sclerosis [14]. The cystic type is also thought to be common in the mandible [15]. We consider our FD case to be (ii) sclerotic form, because the CT imaging features show a frosted glass-like hyperostosis image in both lesions (Figure 2). CT is particularly useful in the diagnosis of this disease and important in confirming the extent of the lesion and determining the treatment plan. In biochemical examination of blood, serum ALP is often elevated, in particular in PFD. As mentioned above, there was no characteristic hematologic finding of elevated ALP in this patient, but CT was most useful in determining the diagnosis and treatment strategy for this patient. Although not performed in this patient, a whole-body CT evaluation to screen for FD-related lesions at other sites should have been considered.

FD in the head and neck region accounts for 10–20% of all cases, with the maxilla accounting for 70% of such cases [16]. FD spreading from the maxilla to the periphery is classified as MFD even if the lesion extends over multiple bone areas [9]. Since the pathogenesis of FD stabilizes as the skeleton matures, osteoplasty or bone loss surgery is often performed for cosmetic purposes after the patient is followed until the end of adolescent growth [17]. However, in rare cases, the disease may progress even after puberty, and surgical treatment is indicated when incidental symptoms or signs of bone thickening appear. In this patient’s case, the development of the lesion had already stopped at the first visit. Although the exact date of onset of this disease is unknown, there have been no active cosmetic complaints due to the disease, and it has not interfered with her daily life for a long period of time. However, the patient had a clear indication for surgery at an advanced age, as she was unable to eat due to a lack of ideal denture fabrication. In craniomaxillofacial FD, since the amount of bone to be removed is determined intraoperatively while comparing with the healthy side of the jaw, it is difficult to grasp the symmetry when only the affected side is developed. In particular, because of the limited field of view in the intraoral approach, even if a 3D model is created and preoperative simulation is performed, sufficient removal is often not possible. For these reasons, the requirement for a surgical support system that allows real-time confirmation of maxillofacial symmetry during surgery remains an unmet need.

The navigation system is a surgical support system that allows intraoperative confirmation of the operation site in three dimensions based on images taken before the operation and has recently been introduced to oral and maxillofacial surgery [18,19,20]. Osteoplasty using a navigation system is also particularly useful for the treatment of FD. By mirroring the healthy side and superimposing it onto the affected side of the jaw, the amount of bone removal can be readily determined intraoperatively while confirming symmetry, thus allowing for a more accurate surgery [21,22,23,24,25]. Wang et al. performed navigation system-based plastic surgery on 13 cases of FD, including cheekbone and maxilla, and reported that the error between preoperative planning and postoperative CT images was less than 2 mm [22]. Gui et al. also reported that the error between pre- and postoperative evaluation of 21 cases of FD performed using the navigation system-based plastic surgery was less than 1 mm [23]. In our case, because the plastic surgery was performed using a navigation system after preoperative mirroring, we were able to remove the bone in a symmetrical manner; even though some asymmetry remained, we could achieve a high level of patient satisfaction and an improved outcome.

The best treatment for FD has not yet been conclusively established [26]. If a patient is diagnosed with FD but it is asymptomatic and there is no evidence of deformity, clinical and imaging follow-up is recommended, with no further surgical intervention. On the other hand, surgical removal is the first choice in symptomatic cases or functional impairment cases due to changes in bone morphology. Most case reports on FD to date have focused on the clinical and radiological features or its surgical management [27,28,29,30,31,32]. However, it should be strongly emphasized that the actual treatment of patients with FD involves not only surgical resection of the affected bone but also prosthetic rehabilitation to restore the patient’s quality of life [33]. Surgical procedures are the mainstay of the overall treatment of FD and are aimed at correcting or preventing functional impairment and achieving normal facial esthetics [28]. To determine the surgical or prosthetic rehabilitation that will improve the patient both esthetically and functionally, an individualized approach should be developed and implemented [33]. In our case, based on the future denture design from the preoperative planning stage, the individualized treatment plan was developed to remove the hyperplastic bone of the maxilla and mandible to correct the occlusal plane and fabricate the denture. Then, early postoperative opening and closing of the mouth training and, thus, stabilizing the jaw position while repairing and adjusting the existing denture was successful in recovering not only oral and systemic health but also comfort, functionality, and esthetics. We believe that comprehensive team management by oral surgeons and maxillofacial prosthetists is essential for the oral rehabilitation of FD patients with dental defects after osteoplasty.

## 4. Conclusions

In the reported case of craniomaxillofacial FD, we were able to obtain a good prognosis for the patient by removing hyperplastic bone using a navigation system and fabricating a denture. Because craniomaxillofacial FD is occasionally associated with bone lesions in other areas, we believe that systemic screening is necessary. In craniomaxillofacial FD, it is necessary to consider the cosmetic and functional deficits and to develop an individualized treatment plan for each patient rather than a routine surgical plan. In addition, we believe that comprehensive team management by oral surgeons and maxillofacial prosthodontists is essential not only for surgery but also for oral rehabilitation after osteoplasty for FD patients with dental defects. Due to the possibility of bone regrowth and rare malignant transformation, we will follow the patient over an extended period.

## Figures and Tables

**Figure 1 diagnostics-12-02146-f001:**
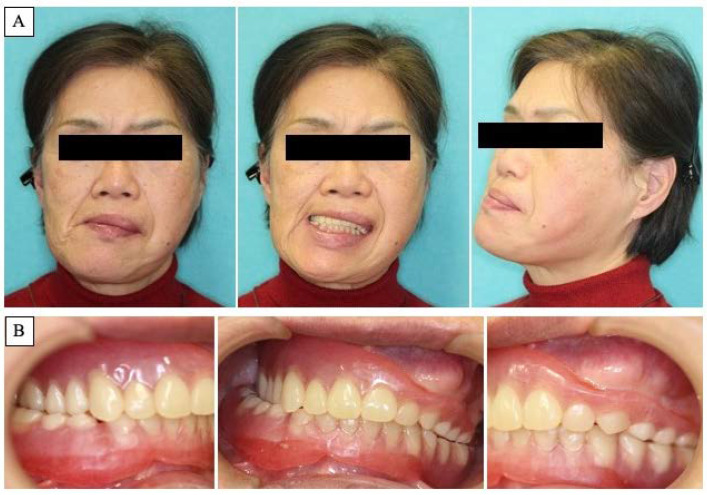
Clinical features at first visit. (**A**): Horizontal inclination of the maxillary occlusal plane and facial asymmetry with right-sided deviation of the mandibular midline. (**B**): Denture with crossbite arrangement at the right side.

**Figure 2 diagnostics-12-02146-f002:**
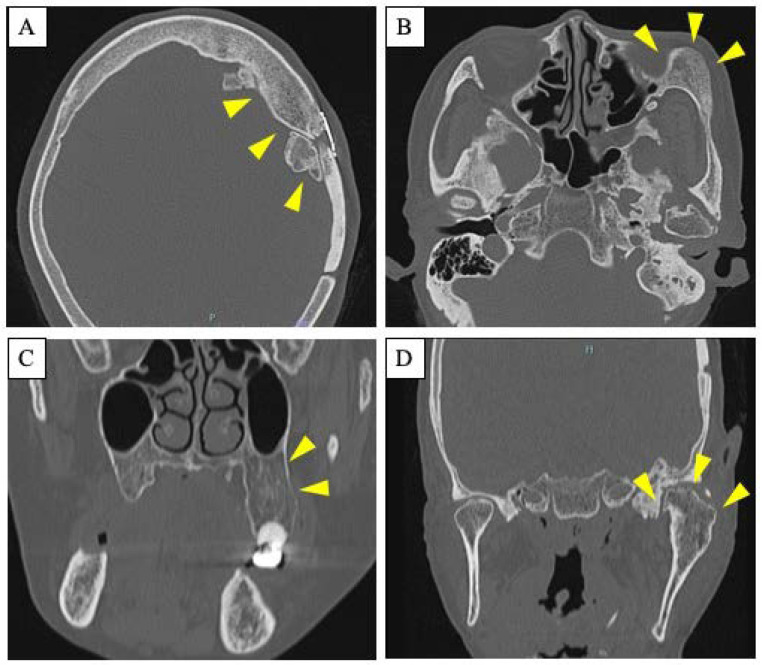
Radiological features. (**A**,**B**): Axial CT scan showing slit-glass-like bone thickening from the left temporal area to the base of the skull (**A**) and zygomatic bone (**B**). Arrowheads indicate the site of the lesion. (**C**,**D**): Coronal CT scan showing bone proliferation in the left alveolar region on the maxilla (**C**) and enlargement of the left articular head on the mandible (**D**). Arrowheads indicate the site of the lesion.

**Figure 3 diagnostics-12-02146-f003:**
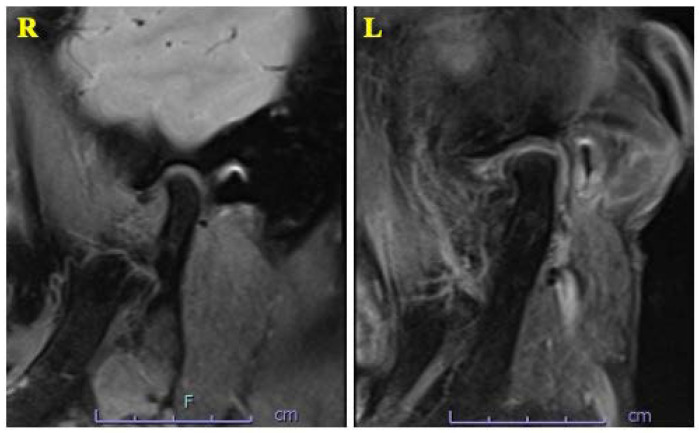
MRI T2-weighted imaging showing irreducible anterior disc displacement of bilateral articular disks and joint effusion in the joint capsule on the left side.

**Figure 4 diagnostics-12-02146-f004:**
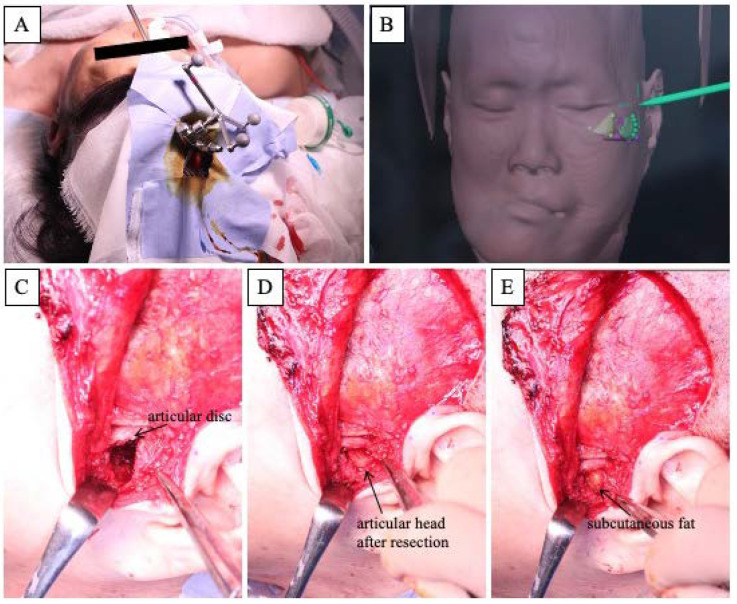
Intraoperative features. (**A**): Landscape of the navigation system. (**B**): Intraoperative screenshot of the navigation system. (**C**–**E**): Approach to the temporomandibular joint area using an Al-Kayat Bramley incision.

**Figure 5 diagnostics-12-02146-f005:**
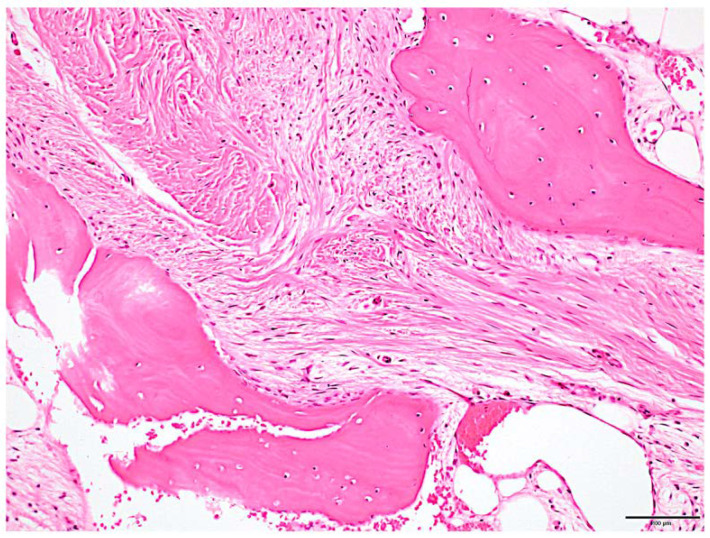
Histologic features. Irregular osteogenic image of bone beam pattern with fibrous connective tissue background was observed. Fibrous bone had poor osteoblast lining. Scale bar: 100 μm.

**Figure 6 diagnostics-12-02146-f006:**
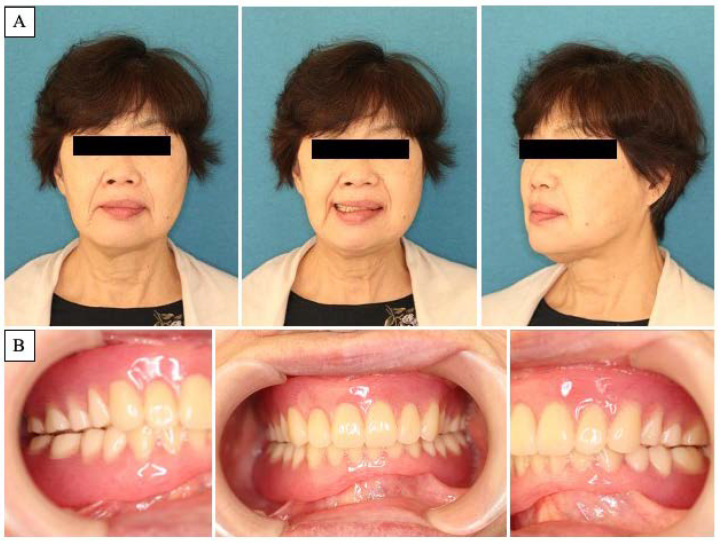
Postoperative clinical features. (**A**): Postoperative facial image. Improvement in facial symmetry is noted. (**B**): Prosthodontist adjusted the denture to achieve a stable jaw position.

**Figure 7 diagnostics-12-02146-f007:**
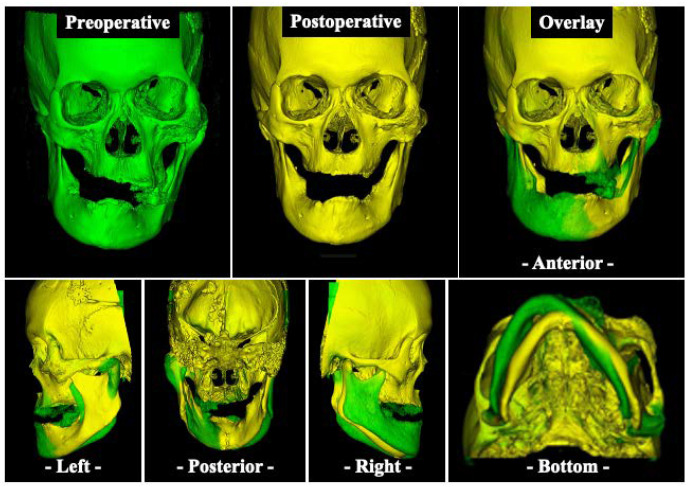
Three-dimensional (3D) based evaluation of craniomaxillofacial bone models. Preoperative (green) and postoperative (yellow) 3D scan of the craniomaxillofacial bone model and overlay (green and yellow) of both scans.

## Data Availability

Not applicable.

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
