# Peer review of "Craniomaxillofacial Fibrous Dysplasia Improved Cosmetic and Occlusal Problem by Comprehensive Treatment: A Case Report and Review of Current Treatments"

_diagnostics, 2022, doi:10.3390/diagnostics12092146_

Round 1

Reviewer 1 Report

This article was a case report of successful surgical treatment for facial fibrous dysplasia.  .

There are some points in the manuscript.

1.    Was the case “polyostotic”? In WHO (Head and neck 4th ed, pp253), “FD occurring in multiple adjacent craniofacial bones is considered to be monoostotic”. So the term “craniofacial fibrous dysplasia” might be precise.

2.    Was the patient “McCune-Albright syndrome”? Was the treatment changed whether the case was McCune-Albright syndrome or not?

3.    Was the patient history, for example hearing loss related to FD?

4.    Histological figure of this FD case is needed. 

5.    In Discussion, line 98, most cases of PFD are sporadic without skin pigmentation. McCune-Albright syndrome and Jaffe-Lichtenstein Syndrome are very rare. Statements should be modified.

Author Response

Reviewer 1

This article was a case report of successful surgical treatment for facial fibrous dysplasia.

Response: Thank you for your review of our paper. We have answered each of your points below.

There are some points in the manuscript.

1. Was the case “polyostotic”? In WHO (Head and neck 4th ed, pp253), “FD occurring in multiple adjacent craniofacial bones is considered to be monoostotic”. So the term “craniofacial fibrous dysplasia” might be precise.

Response: Thank you for your suggestion. Following your suggestion of "Head and neck 4th ed" and "Thoma K, 1954", we have decided to treat our case as "Craniofacial FD". We have reflected this fact in the first paragraph of our Discussion.

2. Was the patient “McCune-Albright syndrome”? Was the treatment changed whether the case was McCune-Albright syndrome or not?

Response: She was not “McCune-Albright syndrome”. However, even if it were "McCune-Albright syndrome," the treatment strategy for craniofacial lesions would have been the same.

3. Was the patient history, for example hearing loss related to FD?

Response: We interviewed the patient and found that she had been experiencing hearing loss since the meningocele surgery and had undergone a tympanostomy. The factual association with FD is unknown, but there have been previous reports of hearing loss from temporal bone FD in the literature, and this may have been a possibility.

4. Histological figure of this FD case is needed.

Response: Thank you for your suggestion. We added the histological figure (Page 2, Line 81, Fig. 5).

5. In Discussion, line 98, most cases of PFD are sporadic without skin pigmentation. McCune-Albright syndrome and Jaffe-Lichtenstein Syndrome are very rare. Statements should be modified.

Response: We thank the reviewer for this comment. We have revised and added to the text (Page 5, Lines 126-127).

Reviewer 2 Report

The submitted manuscript describes the treatment of a case of fibrous dysplasia together with a bibliographic review of the subject.

In general, the presentation is adequate. However, some comments and recommendations are made considering each section.

Introduction

The introduction is correct and concise. Please add a justification about the importance of this clinical case presentation.

Case Report

The photographic material presented to support the description of the case is excellent.

In line 44, is the concept of "horizontal inclination" correct to represent the problem?

In particular, the use of The CARE Guidelines is encouraged:

Gagnier JJ, Kienle G, Altman DG, Moher D, Sox H, Riley D; CAREGroup*. The CARE Guidelines: Consensus-based Clinical Case Reporting Guideline Development. Glob Adv Health Med. 2013 Sep;2(5):38-43. doi: 10.7453/gahmj.2013.008. PMID: 24416692; PMCID: PMC3833570.

In the CARE guideline, among other aspects, the patient's authorization is requested to use information related to their treatment. Therefore, please consider placing the patient's informed consent regarding the use of the information.

Discussion

The first paragraph describes aspects such as the classification of PFD, staging, some aspects of the progress of the pathology, and the potential transformation into a malignant lesion. The authors should discuss how this information relates to the findings observed in the clinical case. The same applies to paragraph 2 related to imaging diagnosis, and paragraph 3, which refers to surgical treatment. This information must not be just a description of the theoretical framework related to pathology.

In paragraph 4, the authors explain why the navigation system is beneficial for treating this type of pathology and support their findings with the literature. Authors are invited to write the previous paragraphs in the same way.

On the other hand, several details in the case description should be addressed in the discussion because they contribute to the knowledge of pathology and its treatment. For example, the patient presented at the age of 63, and I suggest discussing the late treatment of the pathology. On the other hand, the patient presented in her medical history with a meningocele and other conditions. Is this related to the condition under study? In the same sense, figure 2 "radiological features," should be discussed with the literature evidence regarding this pathology aspect.

In lines 153 and 154: "If the patient is asymptomatic and there is no evidence of deformity, clinical and imaging follow-up is recommended, with no further surgical intervention".

I recommend adding to this sentence "If the patient is diagnosed by FD, but it is asymptomatic… .."

Conclusions

In the conclusions, in addition to what has already been described, the authors must add their conclusions from the literature review.

Author Response

Reviewer 2.

The submitted manuscript describes the treatment of a case of fibrous dysplasia together with a bibliographic review of the subject. 

In general, the presentation is adequate. However, some comments and recommendations are made considering each section.

Response: We wish to express our appreciation to the reviewer for insightful comments on our paper. The comments have helped us significantly improve the paper.

Introduction

The introduction is correct and concise. Please add a justification about the importance of this clinical case presentation.

Response: Thank you for your suggestion. We have added the purpose of this case report (Page 1, Lines 37-39).

Case Report

The photographic material presented to support the description of the case is excellent.

Response: Thank you for your kind words.

In line 44, is the concept of "horizontal inclination" correct to represent the problem?

Response: Thank you for pointing this out. We have changed the wording (Page 2, Lines 61-62).

In particular, the use of The CARE Guidelines is encouraged:

Gagnier JJ, Kienle G, Altman DG, Moher D, Sox H, Riley D; CAREGroup*. The CARE Guidelines: Consensus-based Clinical Case Reporting Guideline Development. Glob Adv Health Med. 2013 Sep;2(5):38-43. doi: 10.7453/gahmj.2013.008. PMID: 24416692; PMCID: PMC3833570.

In the CARE guideline, among other aspects, the patient's authorization is requested to use information related to their treatment. Therefore, please consider placing the patient's informed consent regarding the use of the information.

Response: Thank you for pointing this out. We ensure that we obtain the informed consent of our patient. We have added a sentence to the "Introduction" (Page 1, Lines 42-43).

Discussion

The first paragraph describes aspects such as the classification of PFD, staging, some aspects of the progress of the pathology, and the potential transformation into a malignant lesion. The authors should discuss how this information relates to the findings observed in the clinical case. The same applies to paragraph 2 related to imaging diagnosis, and paragraph 3, which refers to surgical treatment. This information must not be just a description of the theoretical framework related to pathology.

In paragraph 4, the authors explain why the navigation system is beneficial for treating this type of pathology and support their findings with the literature. Authors are invited to write the previous paragraphs in the same way.

Response: Thank you for your suggestion. We have added a discussion of our case in each paragraph (Page 5, Lines 132-140; 150-152; 160-162; 164-168;Page 5-6,  176-186).

On the other hand, several details in the case description should be addressed in the discussion because they contribute to the knowledge of pathology and its treatment. For example, the patient presented at the age of 63, and I suggest discussing the late treatment of the pathology. On the other hand, the patient presented in her medical history with a meningocele and other conditions. Is this related to the condition under study? In the same sense, figure 2 "radiological features," should be discussed with the literature evidence regarding this pathology aspect.

Response: Thank you for your kind opinions. We have added a note on the significance of surgery in older FD (Page 5-6, Lines 176-186). There is no association with this FD with regard to known meningeal aneurysms. We have added to our discussion of the CT imaging findings in Figure 2 (Page 5, Lines 160-162).

In lines 153 and 154: "If the patient is asymptomatic and there is no evidence of deformity, clinical and imaging follow-up is recommended, with no further surgical intervention".

I recommend adding to this sentence "If the patient is diagnosed by FD, but it is asymptomatic… .."

Response: Thank you for your suggestion. We have added the sentence you provided (Page 6, Lines 209-210).

Conclusions

In the conclusions, in addition to what has already been described, the authors must add their conclusions from the literature review.

Response: Thank you for your suggestion. We have detailed the conclusions of this paper with more specificity (Page 6-7, Lines 233-245).